# GraphKAN: Graph Kolmogorov Arnold Network for Small Molecule-Protein Interaction Predictions

**Tashin Ahmed** [1]   **Md Habibur Rahman Sifat** [2]

## Abstract

This study presents a proof of concept for utilizing Graph Kolmogorov Arnold Networks (GraphKAN/GKAN) in predicting the binding affinity of small molecules to protein targets. Working with three protein targets, we explored the potential of GraphKAN to infer binding affinities. We compared the performance of GraphKAN with MLP-based graph neural networks, 1D convolutional neural networks (1D CNN), and machine learning algorithms like random forests. Although the model did not achieve state-of-the-art performance, our results demonstrate its feasibility and highlight its promise as a novel approach in computational drug discovery. This work opens new research directions, suggesting that further refinement and exploration of GraphKAN could significantly impact the efficiency and accuracy of binding affinity predictions, ultimately aiding in the discovery of new therapeutic agents. Source code is available at - https://github.com/TashinAhmed/ferroin.

## 1. Introduction

Kolmogorov Arnold Networks (KAN) (Liu et al., 2024), a newly designed replacement of Multi-Layer Perceptron (Haykin, 1998; Cybenko, 1989; Hornik et al., 1989), which has learnable activation functions on edges (weights) instead of fixed activation function on nodes (neurons). Also, research like Liquid Time-Constant Networks (LTCs) (Hasani et al., 2021), a new class of time-continuous recurrent neural networks with modulated linear dynamics and interlinked nonlinear gates, which demonstrate stable behavior, enhanced expressivity, and superior performance in time-series prediction tasks compared to traditional and modern RNNs. These studies are pioneering new directions in AI research, prompting us to explore innovative approaches to developing neural networks.

In this paper, we have prepared a proof of concept employing KAN instead of MLP with Graph Neural Networks (GNN) (Zhou et al., 2020) on a small molecule-protein interaction prediction problem. We have prepared a comparative analysis on the results of a popular machine learning algorithm, i.e., Random Forests (Breiman, 2001), 1 Dimensional CNN (Kiranyaz et al., 2021), a simple MLP-based GNN, and KAN-based GNN.

The search for effective small molecule drugs, which interact with cellular proteins to alter their functions and are crucial for treating various diseases, traditionally involves the laborious and time-consuming process of physically synthesizing and testing molecules against protein targets; given the vastness of the drug-like chemical space, estimated at $10^{60}$ compounds (Kirkpatrick & Ellis, 2004), this method proves impractical for thorough exploration. To address this challenge, researchers tested 133 million small molecules against three protein targets using DNA-encoded chemical library (DEL) technology (Gironda-Martínez et al., 2021), resulting in the creation of the Big Encoded Library for Chemical Assessment (BELKA), an invaluable dataset poised to revolutionize small molecule binding prediction through machine learning (ML). Recent advancements in ML and the availability of large datasets provide an opportunity to revolutionize this process. By leveraging computational models, it is possible to infer the binding affinities of small molecules to protein targets, significantly accelerating the drug discovery process. The goal is to develop models to predict which drug-like small molecules will bind to specific protein targets, thereby paving the way for more accurate and efficient drug development. Related works are presented in **Appendix C**.

For our proof of concept, we utilized a subset of the dataset for initial testing, the results of which are detailed in this manuscript. Future work will involve analyzing the complete dataset.

[1]Independent Researcher, Dhaka, Bangladesh [2]The Hong Kong Polytechnic University, Hong Kong. Correspondence to: Tashin Ahmed <tashinahmed@aol.com>, Md Habibur Rahman Sifat <habib.sifat@connect.polyu.hk>.

*Accepted at the 1st Machine Learning for Life and Material Sciences Workshop at ICML 2024.* Copyright 2024 by the author(s).

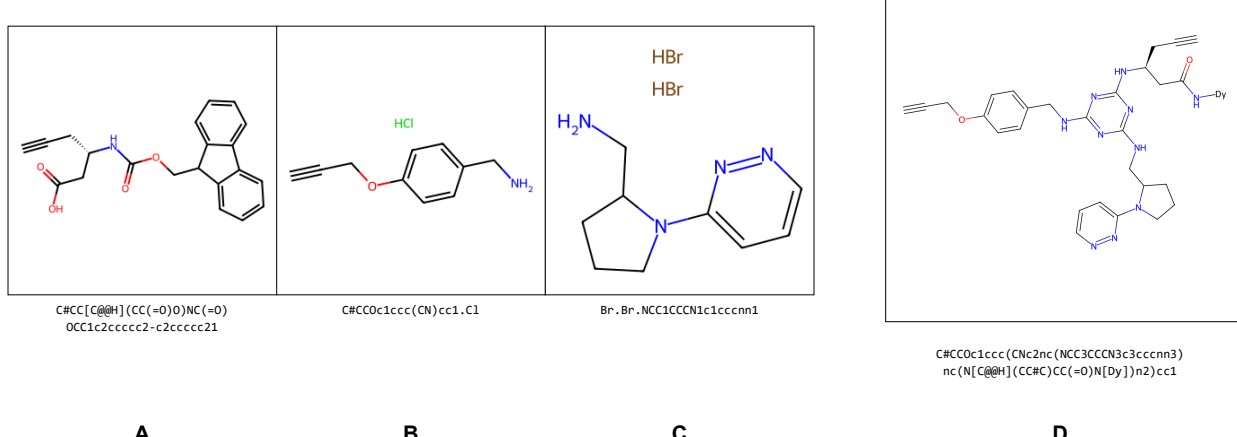

*Figure 1.* A row from the training dataset. A, B, and C are the building blocks in SMILES format. D is the structure of the fully assembled molecule in SMILES. This includes the three building blocks and the triazine core. Note [Dy] stands for the DNA linker. Structures are drawn with rdkit (Bento et al., 2020). For this particular example, the target protein name BRD4 and the binding will not be possible in this case.

## 2. Dataset

The training dataset consists of roughly 98 million examples per protein, with 200,000 validation examples per protein and 360,000 test molecules per protein (Andrew Blevins, 2024). The test set contains building blocks that do not appear in the training set to assess the generalizability of the models. These datasets are highly imbalanced, with approximately $0.5\%$ of examples classified as binders. The data collection involved three rounds of triplicate selection to identify binders experimentally. In our PoC, we have utilized a subset of this large dataset consisting of 20,000 samples per protein target.

### 2.1. Protein Targets

Three protein targets were screened in this study:

- EPHX2 (sEH): Soluble epoxide hydrolase, a potential drug target for high blood pressure and diabetes progression.

- BRD4: Bromodomain 4, involved in cancer progression and targeted by drugs inhibiting its activities.

- ALB (HSA): Human serum albumin, the most common protein in the blood, plays a role in drug absorption and transport.

Details on the dataset preparation are available in **Appendix: A**.

## 3. Methods

### 3.1. Adaption of GNN to Incorporate KAN

In this study, we showed the adaptation of the traditional GNN by integrating KANs. This approach aims to enhance the feature transformation capabilities of the GNN, leveraging the learnable activation functions of KANs for improved expressivity and performance.

#### 3.1.1. GNN ARCHITECTURE

The baseline GNN model utilized in our study consists of multiple layers, each performing the following steps:

- **Node Feature Aggregation:** Node features and edge attributes are combined using max aggregation.

- **Feature Update:** A linear layer updates the aggregated node features.

- **Non-linear Transformation:** Batch normalization, ReLU activation, and dropout are applied sequentially to the updated features.

- **Global Pooling:** After processing through all GNN layers, global max pooling aggregates node features into a graph-level representation.

- **Output Generation:** The pooled features are passed through a final linear layer to produce the output.

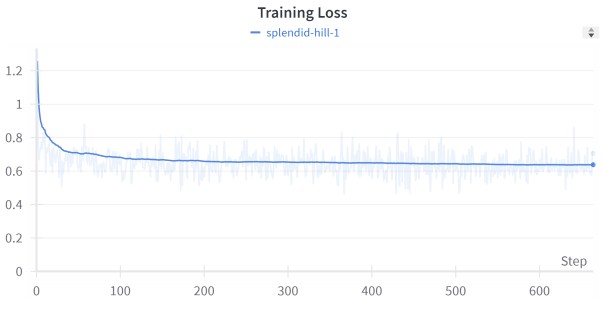

*Figure 2.* GraphKAN: example of training loss for 10 epochs on sEH.

### 3.1.2. INCORPORATION OF KAN

We replace the standard MLP in the GNN with the Naive Fourier KAN layer. The KAN layer's forward pass involves Fourier transformations, enhancing the model's ability to capture complex patterns in the data.

**Initialization (Algorithm 1):** Input dimensions and output dimensions are defined. Fourier coefficients are initialized, sampled from a normal distribution, and scaled appropriately. If bias addition is enabled, a bias term is initialized.

**Forward Pass (Algorithm 2):** Input data is reshaped and Fourier-transformed. Cosine and sine components are computed and concatenated. Fourier-transformed features are processed through Einstein summation to yield the output, with an optional bias addition.

### 3.1.3. INTEGRATION WITH GNN

The Naive Fourier KAN layer is integrated into the GNN layers as follows:

Each GNN layer combines node features and edge attributes before applying the Fourier transformation using the KAN layer. This integration enhances feature transformation, enabling the model to capture more intricate relationships within the graph data.

### 3.2. Proposed GraphKAN

In GraphKAN, we have replaced the MLP with KAN to create a simple GNN. We have used Naive Fourier Layer as the proposed architecture. The reasoning behind choosing the layer has been discussed in **section 4**.

Naive Fourier KAN Layer has been presented in **Algorithm 1** and the forward pass of the layer in **Algorithm 2**.

### 3.3. Evaluation Metric

Evaluation script computes the average precision score for different subsets of the dataset and then averages those scores. Which means the average precision for each protein and each split group is calculated individually and then been averaged. This method ensures a balanced evaluation across different groups.

$$mean\_score = \frac{1}{|score|} \sum_{i=1}^{|N_p|} \sum_{j=1}^{|S_g|} \quad (1)$$
$$(1_{|select_{ij}|>0}) \cdot score_{ij}$$

In **equation 1**, $protein\_names$ ($N_p$) and $split\_groups$ ($S_g$) are the lists of unique protein names and split groups. $1_{|select_{ij}|>0}$ is an indicator function that is 1 if the subset $select_{ij}$ is not empty and 0 otherwise. If the subset is not empty $if \ |select_{ij}| > 0$, then the subset of rows where the protein name and split group match is defined as $score_{ij}$.

Details on the evaluation metric are discussed in **Appendix B**.

## 4. Experiments and Discussions

We have experimented with a subset of the dataset with 20,000 samples of each protein and tried to provide a comparison among Random Forests, vanilla 1D CNN, a simple MLP-based GNN, and our proposed KAN-based GNN.

To generate ECFP features, we utilized RDKit, an open-source cheminformatics tool, which efficiently creates hashed bit vectors, streamlining the feature generation process for robust molecular representation in machine learning applications (Bento et al., 2020).

---

**Algorithm 1** Naive Fourier KAN Layer Initialization

1: **Input:**
2:    input_dim: Dimension of the input data
3:    out_dim: Dimension of the output data
4:    grid_size: Number of grid points for Fourier features (default: 300)
5:    add_bias: Boolean flag to indicate if bias should be added (default: True)
6: Initialize Fourier coefficients **F** with shape $(2, out\_dim, input\_dim, grid\_size)$, sampled from a normal distribution and scaled by $\frac{1}{\sqrt{input\_dim} \cdot \sqrt{grid\_size}}$
7: **if** add_bias is True **then**
8:    Initialize bias **b** with shape $(1, out\_dim)$
9: **end if**

---

---

**Algorithm 2** Naive Fourier KAN Layer Forward Pass

---

1: **Input:** $\mathbf{x}$ of shape $(N, \text{input\_dim})$
2: Reshape $\mathbf{x}$ to $(N, \text{input\_dim})$
3: Construct grid vector $\mathbf{k}$ as $\mathbf{k} = [1, 2, \dots, \text{grid\_size}]$
4: Reshape $\mathbf{x}$ to $(N, 1, \text{input\_dim}, 1)$
5: Compute cosine component $\mathbf{C}$ as $\mathbf{C} = \cos(\mathbf{k} \cdot \mathbf{x})$
6: Compute sine component $\mathbf{S}$ as $\mathbf{S} = \sin(\mathbf{k} \cdot \mathbf{x})$
7: Reshape $\mathbf{C}$ and $\mathbf{S}$ to $(1, N, \text{input\_dim}, \text{grid\_size})$
8: Concatenate $\mathbf{C}$ and $\mathbf{S}$ along the first dimension, resulting in shape $(2, N, \text{input\_dim}, \text{grid\_size})$
9: Perform the Einstein summation $\mathbf{y} = \text{einsum}(\mathbf{F}, [d, b, i, k], \mathbf{CS}, [d, j, i, k] \rightarrow b, j)$, yielding shape $(N, \text{out\_dim})$
10: **if** add\_bias is True **then**
11:   Add bias $\mathbf{b}$ to $\mathbf{y}$
12: **end if**
13: Reshape $\mathbf{y}$ to the original output shape
14: **Output:** Return the computed output $\mathbf{y}$

---

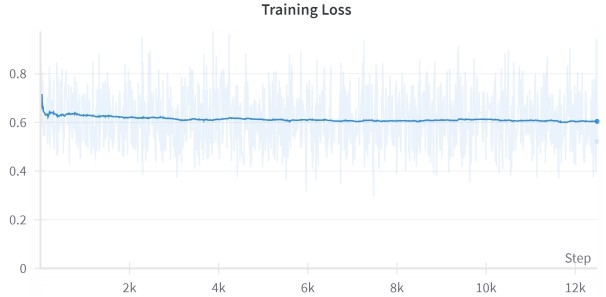

*Figure 3.* GraphKAN: example of training loss for 100 epochs on sEH.

### 4.1. Comparative Evaluation with Other Approaches

We have used sklearn RandomForestClassifier (Varoquaux et al., 2015) with 100 trees to evaluate the performance using Extended-Connectivity Fingerprints (ECFPs). Despite its simplicity, this technique has demonstrated efficacy, often performing comparably to more advanced methods in prior benchmarks (Weinberger et al., 2009; Rogers & Hahn, 2010). The process involves decomposing molecular graphs into collections of subgraphs, which are then hashed into bit vectors, creating a compact representation of the molecular structure. This is analogous to the hashing trick used in NLP for representing bags of words before transformer models (Toure et al., 2013; Karthikeyan et al., 2014).

A simple 1D-CNN is trained for 100 epochs and the highest dimension of 1024.

We trained a vanilla GNN model using PyTorch Geometric (Fey & Lenssen, 2019). The GNN Layer combines node features and edge attributes, using max aggregation and a linear layer to update node features. The model consists of 6 such layers, each followed by batch normalization, ReLU activation, and dropout (0.3 probability). After processing through these layers, global max pooling aggregates node features into a graph-level representation, which is passed through a final linear layer to produce the output. The model's current scores shown in **Table 1** are trained using the AdamW optimizer with a learning rate of 0.001 and a binary cross-entropy loss function. Training is conducted over 100 epochs, with a hidden dimension size of 128 and a batch size of 32.

### 4.2. Analysis on GraphKAN

GraphKAN incorporates a naive Fourier KAN layer to enhance feature transformations. The NaiveFourierKANLayer applies Fourier transformations to input features, using a grid size of 300 (Liu et al., 2024) and adding a bias term. This layer is used within the GNN Layer, which combines node features and edge attributes before applying the Fourier transformation. The GNN model consists of 6 such custom layers, each followed by batch normalization and a dropout rate of 0.3 for the preliminary experimentations.

It has been observed that implementing the KAN for graphs in the latent feature space significantly enhances performance. Specifically, employing a linear layer to project the input features into a latent space prior to applying the KAN layer is crucial. This can be achieved by using a linear transformation such as:

```
self.lin_in = nn.Linear(in_feat,
hidden_feat, bias=use_bias)
```

This approach has been found to be superior compared to directly applying the KAN layer on the input features, which can be represented as:

```
self.lin_in = KANLayer(in_feat,
hidden_feat, grid_feat,
addbias=use_bias)
```

Notably, without the linear layer for low-dimensional latent feature projection, the KAN layer lacks a training signal, impeding its ability to train effectively. This intriguing observation suggests a need for further theoretical exploration following empirical validation. Whether this technique is equally effective for other data types, such as images and text, remains to be determined.

In our experiments, the Stochastic Gradient Descent (SGD) optimizer and its variant ASGD (Bottou, 2012) demonstrated more excellent stability compared to the Adam and

*Table 1.* Outcomes of the experimented approaches on a subset of the dataset with 100 iterations each. The average precision for each protein and each split group will be calculated individually, and these scores will then be averaged. Detailed explanation on the evaluation metric described in **Appendix B**.

| MODEL | MEAN SCORE |
|---|---|
| RANDOM FORESTS | 0.251 |
| 1D CNN | 0.402 |
| VANILLA GNN | 0.193 |
| GRAPHKAN | **0.428** |

AdamW optimizers (Zhuang et al., 2022). However, a trade-off was observed in terms of convergence speed. Adam converged in approximately 300 epochs, whereas SGD required around 12,000 epochs to achieve similar results. Despite the slower convergence, the stability of SGD might be advantageous in certain scenarios.

## 5. Future Works

As KAN has primarily been implemented and demonstrated on small datasets (Liu et al., 2024), it presents a significant challenge to validate its efficacy and performance on larger datasets such as BELKA, which exceeds 50GB of parquet files. We discussed about the motivation behind implementing KAN Protein-Ligand Bioaffinity predictions in **Appendix D**. Currently, we are employing a batch-by-batch training approach to process at least 50% of the dataset. Future work will focus on refining the manuscript to include detailed architectural explanations and the rationale behind the selection of network structures, providing a more comprehensive understanding of KAN's application and potential in large-scale data scenarios. We will also open-source the source codes, models, and detailed explanations after the end of the competition in a repository that will be made available following the initial review process. After the initial review, we will only open-source the primary network. Once the competition concludes, we will release the full codebase, including the weight files. Also, we have to enhance the words of the manuscript, fix typos, and arrange the sections properly.

GraphKAN has the potential to replace large networks by building solutions with comparatively small architectures, making them more viable in terms of cost and business applications. Our work represents a small step towards achieving this goal. We have discussed commercial potential of GraphKAN in **Appendix E**.

We have also ensembled the models outcome to boost the overall performance over the dataset. The weights distributions for the outcome is $0.436$ are GraphKAN: 0.71, Random Forests: 0.07, GNN: 0.05, 1D CNN: 0.17 to improve the outcome for the competition.

## 6. Conclusion

In this study, we introduced GraphKAN to predict the binding affinity of small molecules to protein targets. By conducting a proof of concept with three protein targets, we demonstrated the feasibility and potential of GraphKAN in this domain. Our comparative analysis with other machine learning models, including MLP-based graph neural networks, 1D convolutional neural networks (1D CNN), and random forests, highlighted that while GraphKAN did not achieve state-of-the-art performance, it shows promise as an innovative method in computational drug discovery. The results of our experiments highlight the need for further refinement and exploration of GraphKAN. Specifically, future work will focus on validating the performance of GraphKAN on larger datasets such as BELKA, which poses significant challenges due to its size and complexity. Additionally, we will delve deeper into the architectural aspects of GraphKAN, providing detailed explanations and rationales behind our design choices. Our findings suggest that GraphKAN could significantly enhance the efficiency and accuracy of binding affinity predictions with continued development. This would ultimately aid in discovering new therapeutic agents by enabling more effective exploration of the vast chemical space. To support the broader research community, we plan to open-source our models and codebase following the completion of the initial review process and the conclusion of the related competition. In conclusion, while this study represents an early step in leveraging GraphKAN for drug discovery, it opens new avenues for research. It highlights the potential of innovative neural network architectures in advancing the field of computational biology. Combining ensemble methods to boost performance, our work sets the stage for more sophisticated and practical approaches to predicting molecular interactions, potentially transforming the drug development landscape.

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
