# A. Dataset Description

Dataset is processed, prepared and collected through the use of DNA-encoded chemical library (DEL) technology, specifically tailored to evaluate the binding affinity of small molecules to 3 protein targets. DELs are used to create large libraries of small molecules with unique DNA barcodes for each molecule. Small molecules in the DEL are combined using a combinatorial approach, allowing the creation of a vast number of unique molecules from a limited set of building blocks. In our case, 3 types of building blocks, combining them can generate a large number of different molecules. The DELs are pooled together, meaning multiple small molecules are tested simultaneously rather than individually. These pools are exposed to the protein targets in solution. The proteins of interest are rinsed to remove non-binding molecules, and the remaining bound molecules are identified through DNA sequencing. The 3 protein targets are used: EPHX2 (sEH), BRD4, and ALB (HSA). Each target has been chosen due to its relevance in various medical conditions and its history of being screened using DEL approaches.

The chemical structure of each small molecule is represented using the Simplified Molecular-Input Line-Entry System (SMILES), a standardized string notation that encodes the molecule's structure. The dataset includes SMILES representations for the fully assembled molecule as well as its constituent building blocks. Each molecule's ability to bind to a protein target is labeled with a binary classification indicating whether it binds (1) or does not bind (0). This binding data is derived from the experimental results of the DEL screening process. The complete dataset is split into training and test sets. The training set includes molecules with known binding outcomes, while the test set is used to evaluate the predictive models (future work). The training data includes around 98 million examples per protein target, while the validation set has around 200,000 examples per protein, and the test set has about 360,000 examples per protein. The dataset is highly imbalanced, with only about 0.5% of the examples classified as binders. This reflects the real-world scenario where only a tiny fraction of tested molecules exhibit binding affinity to the target proteins.

## A.1. Chemical Representations

One goal of this research is to explore various ways of representing molecules. Small molecules can be represented using SMILES, graphs, 3D structures, and other methods. This study provides molecules in SMILES format, which encodes the molecular graph, including atoms, bonds, connectivity, and stereochemistry as a linear sequence of characters. SMILES is widely used in ML applications for chemistry due to its standardized and machine-readable format.

## A.2. Experimental Details

DEL technology allows the efficient testing of a vast number of small molecules by attaching unique DNA barcodes to each molecule. This approach enables the pooling of many molecules in a single tube, where they are exposed to the protein target of interest. After rinsing away non-binders, the remaining binders are identified through DNA sequencing. DELs are created using a combinatorial approach, significantly expanding the library of small molecules from a limited number of building blocks.

The dataset for this study includes examples represented by binary classifications indicating whether a given small molecule binds to one of three protein targets. The data were collected using DEL technology and are provided in both CSV and Parquet formats. The dataset includes the following features:

- id: A unique identifier for the molecule-binding target pair.

- buildingblock1_smiles, buildingblock2_smiles, buildingblock3_smiles: The structures of the three building blocks in SMILES format.

- molecule_smiles: The structure of the fully assembled molecule in SMILES format.

- protein_name: The name of the protein target.

- binds: A binary class label indicating whether the molecule binds to the protein (not available for the test set).

## A.3. Data Composition

The training dataset consists of roughly 98 million examples per protein, with 200,000 validation examples per protein and 360,000 test molecules per protein. The test set contains building blocks that do not appear in the training set to test the generalizability of the models. These datasets are highly imbalanced, with approximately 0.5% of examples classified as binders. The data collection involved three rounds of selection in triplicate to identify binders experimentally.

## A.4. Protein Targets

Three protein targets were screened in this study:

- EPHX2 (sEH): Soluble epoxide hydrolase, a potential drug target for high blood pressure and diabetes progression.

- BRD4: Bromodomain 4, involved in cancer progression and targeted by drugs inhibiting its activities.

- ALB (HSA): Human serum albumin, the most common protein in blood, playing a role in drug absorption and transport.

These targets were chosen by the organizers to provide a diverse set of challenges and to evaluate the potential of ML models to generalize across different types of protein interactions.

### A.5. Data Splitting and Scoring

The dataset was split using multiple techniques to ensure a comprehensive evaluation of model performance. Reserved building blocks ensured that certain molecules only appeared in the test set, while random and scaffold-based separations were also employed. The scoring metric calculates the average precision across different groups to ensure a balanced evaluation.

By leveraging BELKA, this study aims to advance ML approaches in small molecule chemistry, facilitating more efficient drug discovery and potentially leading to new life-saving medicines.

## B. Evaluation Metric Description

The dataset was split using various techniques to ensure diversity and avoid overfitting to specific types of molecules, which included keeping reserved building blocks for the test set and ensuring they did not appear in the training set. Specific Murcko scaffolds were made exclusive to the test set, then additionally sampled random molecules. A property chemical library was included solely in the test set. A single AP value as the metric inadvertently emphasized understanding base rates. The differences in base rates and difficulty levels across splits and proteins led to an imbalanced evaluation. To ensure fair and balanced evaluation, the proposed metric calculates AP for each protein and each split group individually and then averages these scores. The method ensures a balanced evaluation across different groups, avoiding bias towards any particular split or protein. This study aims to identify molecule representations and model architectures using GraphKAN that provide a formidable generalization across different chemical spaces and conditions. Including a new library in the test set ensures the models are tested on out-of-distribution molecules, promoting the development of models that generalize well. AP was calculated across 9 groups (shared and non-shared building blocks plus a new library for 3 proteins) and averaged, with 2/3 of the final score coming from molecules outside the training distribution.

The step-by-step procedure of the evaluation follows,

Initialized the lists of unique protein names and split groups, $protein\_names$ ($N_p$) and $split\_groups$ ($S_g$). Both of them consist of unique protein names from the solution data frame. Loop through each combination of protein name and split group: For each $protein\_name_i \in N_p$ and $split\_group_j \in S_g$:

Selected the subset of rows where the protein name and split group match:

$$select_{ij} = k \mid solution[k, 'N_p'] = protein\_name_i \\ and\ solution[k, 'S_g'] = split\_group_j \quad (2)$$

Check if the subset is not empty $if\ |select_{ij}| > 0$:

$$score_{ij} = average\_precision\_score( \\ solution[k, target] \mid k \in select_{ij}, \quad (3) \\ submission[k] \mid k \in select_{ij})$$

Compute the mean of the collected scores:

$$mean\_score = \frac{1}{|score|} \sum_{score\ \in\ scores} score \quad (4)$$

So, the overall evaluation equation can be written as,

$$mean\_score = \frac{1}{|score|} \sum_{i=1}^{|N_p|} \sum_{j=1}^{|S_g|} \\ (1_{|select_{ij}|>0}) \cdot score_{ij} \quad (5)$$

Here $1_{|select_{ij}|>0}$ is an indicator function that is 1 if the subset $select_{ij}$ is not empty and 0 otherwise.

## C. Related Works

*1. DeepChem: An Open-Source Toolbox for Deep Learning in Drug Discovery* (Ramsundar et al., 2019)

An open-source toolkit that provides a suite of DL algorithms tailored for drug discovery and materials science. The authors demonstrate tasks such as molecular property prediction, molecular generation, and protein-ligand binding affinity prediction. The toolkit integrates several SOTA DL models and emphasizes the importance of model Interpretability and robustness in drug discovery applications.

Provides a comprehensive deep learning toolbox for cheminformatics. Demonstrates the effectiveness of deep learning in predicting molecular properties and binding affinities. Emphasizes the importance of reproducibility and open science in drug discovery research.

The models may require significant computational resources, making them less accessible for smaller labs. DL models

can be difficult to interpret, posing challenges in understanding how predictions are made. Performance is highly dependent on the quality and quantity of training data.

*2. Predicting Drug-Target Interactions Using Restricted Boltzmann Machines* (Wang et al., 2014)

In this study, the authors employ Restricted Boltzmann Machines (RBMs) to predict drug-target interactions (DTIs). The model is trained on known DTIs and then used to identify potential new interactions. The paper demonstrates that RBMs can effectively capture the complex relationships between drugs and their targets, outperforming traditional methods such as similarity-based approaches.

Introduces RBMs as a powerful tool for DTI prediction. Demonstrates superior performance over traditional similarity-based methods. Provides insights into the potential of deep learning techniques in understanding drug-target relationships.

RBMs require careful tuning of hyperparameters, which can be time-consuming. They may not scale well with vast datasets. The model's performance can degrade with noisy or incomplete data.

*3. Machine Learning on DNA-Encoded Libraries: A New Paradigm for Hit Finding* (McCloskey et al., 2020)

This paper explores the use of machine learning to analyze data from DELs, a technology that allows for the simultaneous screening of billions of compounds against a target protein. The authors develop machine learning models to predict the binding affinities of compounds in DELs, showing that these models can accurately identify promising drug candidates from vast chemical spaces.

Utilizes machine learning to process and analyze DEL data. Demonstrates the ability to predict binding affinities with high accuracy. Highlights the potential of combining DEL technology with ML to accelerate drug discovery.

The approach relies heavily on the quality and comprehensiveness of the DEL data. ML models may struggle with generalizing novel chemical spaces not represented in the training data. Interpretability of the models remains a challenge, especially for complex biological interactions.

*4. Generative Models for Molecular Discovery: Recent Advances and Challenges* (Elton et al., 2019)

This review paper surveys recent advancements in using generative models for molecular discovery. It discusses various generative techniques, including VAEs, GANs, and RL, in the context of drug discovery. The authors highlight successful applications, such as generating novel drug-like molecules and optimizing molecular properties.

Provides a comprehensive review of generative models in drug discovery. Highlights successful applications and case studies. Discusses the challenges and future directions for generative model research in drug discovery.

Generative models often require large datasets for training, which may not always be available. These models can be computationally intensive and require significant resources. Ensuring the chemical validity and synthesizability of generated molecules remains a challenge.

*5. Deep Learning for Drug Discovery and Biomarker Development* (Vamathevan et al., 2019)

This paper reviews the application of deep learning techniques in drug discovery and biomarker development. It covers various deep learning models, such as CNNs and RNNs, and their applications in predicting drug efficacy, toxicity, and mechanism of action. The review also discusses the integration of multi-omics data and the potential of deep learning to uncover new biomarkers for disease.

Reviews the application of deep learning in drug discovery and biomarker development. Discusses the integration of multi-omics data for comprehensive analysis. Highlights the potential of deep learning to improve drug efficacy and safety predictions.

DL models can be data-hungry, requiring large, annotated datasets for training. Interpretation of model predictions remains a challenge, particularly in understanding biological relevance. Integrating and harmonizing multi-omics data can be complex and resource-intensive.

## D. Motivation for Using GraphKAN

The motivation for using GraphKAN or KAN in general in protein-ligand bioaffinity predictions stems from their theoretical robustness, enhanced expressivity, and improved feature transformation capabilities (**Table 2**). While there are potential disadvantages, such as increased computational complexity and risk of overfitting, the overall benefits make KANs a promising approach for this domain. Proper implementation and regularization can address these challenges, leading to more accurate and reliable bioaffinity predictions.

## E. Commercial Potential of GraphKAN

Firstly, GraphKAN itself represents an innovative approach in drug discovery. By utilizing GraphKAN, the research enhances the feature transformation capabilities of GNN through the integration of learnable activation functions. This novel method is designed to improve the prediction of binding affinities between small molecules and protein targets, a critical aspect of drug discovery. Although the model has not yet achieved state-of-the-art performance, its potential for refinement suggests it could significantly

| Advantages | Potential Disadvantages |
| --- | --- |
| **1. Theoretical Foundation:** KANs are grounded in the Kolmogorov-Arnold representation theorem, which states that any multivariate continuous function can be represented as a finite sum of continuous functions of one variable. This theoretical underpinning provides a robust framework for learning complex, non-linear relationships, making KANs highly suitable for tasks involving intricate data structures, such as protein-ligand interactions. | **1. Computational Complexity:** The integration of KANs increases the computational complexity of the model. The Fourier transformations and the learning of multiple coefficients add to the computational overhead, potentially leading to longer training times and higher resource consumption. This trade-off between accuracy and computational efficiency must be carefully managed. |
| **2. Enhanced Expressivity:** The key advantage of KANs lies in their ability to learn more expressive activation functions compared to traditional NN architectures. KANs can adaptively capture complex patterns in the input data by incorporating learnable Fourier coefficients. This is particularly beneficial for protein-ligand bioaffinity predictions, where the interactions are inherently complex and multi-faceted. | **2. Overfitting Risks:** Given the high expressivity of KANs, there is a potential risk of overfitting, especially when dealing with small datasets. The model might learn noise in the training data rather than the underlying patterns, adversely affecting its generalization to unseen data. Regularization techniques and careful model validation are necessary to mitigate this risk. |
| **3. Improved Feature Transformation:** Incorporating KANs into a GNN enhances the feature transformation process. The Fourier transformation within the KAN layer allows the model to better capture periodicities and other intricate patterns in the molecular data. This improved feature transformation leads to more accurate and robust predictions of bioaffinity. | **3. Complexity in Implementation:** The implementation of KANs requires a more complex architecture compared to standard neural networks. This added complexity might pose challenges regarding model design, debugging, and maintenance. Ensuring that the benefits outweigh the implementation challenges is critical. |
| **4. Flexibility and Adaptability:** KANs provide flexibility in learning activation functions tailored to the specific data and task at hand. This adaptability is crucial in the context of protein-ligand bio affinity predictions, where different types of interactions (e.g., hydrophobic, hydrogen bonding, van der Waals forces) need to be accurately modeled. | **4. Hyperparameter Sensitivity:** KANs require careful tuning of hyperparameters such as the number of layers, the number of Fourier coefficients, and the learning rates, as far as we have discovered from the literature. As a novel system for NNs, more trial and error is needed to understand this architecture's behavior on different data sets. This process can be time-consuming and computationally intensive, posing a challenge in achieving the best model performance. |

*Table 2.* The motivation for using KANs in protein-ligand bioaffinity predictions lies in their theoretical robustness, enhanced expressivity, and improved feature transformation. Despite potential disadvantages like increased computational complexity and overfitting risk, KANs offer significant benefits. Proper implementation and regularization can mitigate these challenges, ensuring accurate and reliable predictions.

improve the efficiency and accuracy of molecular interaction predictions.

Secondly, the commercial viability of this research is evident in its potential to reduce costs and time associated with traditional drug discovery methods. Conventional approaches often involve extensive physical synthesis and testing, which are both time-consuming and expensive. Computational models like GraphKAN can dramatically cut down on these costs and time requirements by predicting binding affinities more efficiently. Moreover, the research leverages the Big Encoded Library for Chemical Assessment (BELKA) dataset, which includes 133 million small molecules tested against three protein targets. This extensive dataset provides a robust foundation for developing accurate predictive models, making the technology commercially appealing for pharmaceutical companies.

The market potential of this research is substantial. The protein targets studied—EPHX2, BRD4, and ALB—are associated with diseases such as high blood pressure, diabetes, and cancer. Accurately predicting interactions with these proteins can lead to the development of new therapeutic agents, opening up significant market opportunities. Additionally, the adaptability of the GraphKAN model to other protein targets and molecular datasets suggests that this technology could be broadly applied across various drug discovery projects, further increasing its commercial potential.

Furthermore, the open-source nature of the project and the encouragement of collaboration enhance its commercial appeal. By making the source code available on GitHub, the researchers facilitate further development and refinement of the technology, which can accelerate its adoption in commercial applications. Future work aims to analyze larger datasets and refine the model, ensuring continuous improvement and increasing the technology's attractiveness to potential commercial partners.

In conclusion, the research on GraphKAN presents a promising new direction in computational drug discovery. Its potential to enhance prediction accuracy and efficiency, combined with its applicability to various therapeutic targets, makes it commercially exciting. The open-source approach and ongoing efforts to refine and scale the model further enhance its appeal to the pharmaceutical industry and other stakeholders in drug development.