# OpenReview forum: "GraphKAN: Graph Kolmogorov Arnold Network for Small Molecule-Protein Interaction Predictions"
_ICML.cc/2024/Workshop/ML4LMS — ML4LMS Poster_

### Official Review · Reviewer_DbRp · 2024-06-11

**Rating:** 3
**Confidence:** 3

**Review:**

### **Summary**

The work presents GraphKAN, an early attempt at utilizing Kolmogorov Arnold Networks (KANs) in place of MLPs in Graph Neural Network (GNNs). The method is assessed on the bioaffinity prediction task.

### **Strengths**

- An early work that explores a newly developed idea, KANs, on the topic of protein-ligand binding affinity.

### **Weaknesses**

- Missing major details surrounding the methods i.e. the adaptation of the GNN to incorporate KANs. Only form of implementation details are presented in Algorithms, but the details should be expanded upon in a proper methods section and should be included in the main manuscript.
- No details regarding the motivation on the use of KANs, and their advantage (also disadvantage if any) on the case of protein-ligand bioaffinity predictions. Essentially, why KANs?
- Poor formatting of the sections, though authors acknowledge this in the Future Works section.


### **Questions**

- What are the limitations of KANs?
- Are there any special tricks that had to be done to ensure good performance?

### **Comments / Suggestions**

- Following sentence sounds out-of-place as it is unrelated to KANs and Protein-Ligand Interactions (Introduction lines 38-46):
    > Also, research like Liquid Time-Constant Networks (LTCs) (Hasani
    et al., 2021), a new class of time-continuous recurrent neural networks with modulated linear dynamics and interlinked nonlinear gates, which demonstrate stable behavior,
    enhanced expressivity, and superior performance in timeseries prediction tasks compared to traditional and modern RNNs.

- Evaluation Metric equation (1) is badly formatted. It might be better to assign long names to variables, and then define the variables i.e |protein_names| → $N_p$ to signify number of unique proteins.
- The work has potential to be more interesting. It would be great if more details surrounding the method were presented. It could also be valuable to evaluate this method against other benchmarks that have been evaluated on by other methods in the fields like CASF-16 and Merck FEP benchmark for bioaffinity prediction.

### **Summary of Review**
The work explores a new alternative to MLPs with KANs in GNNs; however, the work is missing crucial details regarding the method with incomplete or lack of sections surrounding this, did not clarify a motivation for using KANs, and it could explore the performance of the network on more bioactivity benchmarks. With this, I recommend a reject.

---

### Official Review · Reviewer_8gFC · 2024-06-11
**New method for an important problem**

**Rating:** 8
**Confidence:** 4

**Review:**

GraphKAN seems to work well. Hope to see more details in the final version.